# Exercise and Worsening of Extrapyramidal Symptoms during Treatment with Long-Acting Injectable Antipsychotics

**DOI:** 10.3390/pharmacy9030123

**Published:** 2021-07-03

**Authors:** David D. Kim, Donna J. Lang, Darren E. R. Warburton, Alasdair M. Barr, Randall F. White, William G. Honer, Ric M. Procyshyn

**Affiliations:** 1Department of Anesthesiology, Pharmacology & Therapeutics, University of British Columbia, Vancouver, BC V6T 1Z4, Canada; davidd.kim@alumni.ubc.ca (D.D.K.); al.barr@ubc.ca (A.M.B.); 2British Columbia Mental Health & Substance Use Services Research Institute, Vancouver, BC V5Z 3L7, Canada; donna.lang@ubc.ca (D.J.L.); william.honer@ubc.ca (W.G.H.); 3Department of Radiology, University of British Columbia, Vancouver, BC V6T 1Z4, Canada; 4School of Kinesiology, University of British Columbia, Vancouver, BC V6T 1Z4, Canada; darren.warburton@ubc.ca; 5Department of Psychiatry, University of British Columbia, Vancouver, BC V6T 1Z4, Canada; randall.white@vch.ca

**Keywords:** antipsychotics, long-acting injectable antipsychotics, paliperidone palmitate, extrapyramidal symptoms, exercise, physical activity, schizophrenia

## Abstract

Second-generation antipsychotic medications are used to treat schizophrenia and a range of other psychotic disorders, although adverse effects, including cardiovascular and metabolic abnormalities and extrapyramidal symptoms, are often inevitable. Studies have shown that exercise, as an adjunct therapy, can be effective in reducing the core symptoms of schizophrenia as well as ameliorating intrinsic and antipsychotic-induced cardiometabolic abnormalities. However, it is noteworthy that exercise may need to be implemented with caution in some individuals receiving certain antipsychotic treatment regimens. We report here two cases of exercise-associated worsening of extrapyramidal symptoms in two individuals with schizoaffective disorder treated with a long-acting injectable antipsychotic medication over the course of a 12-week exercise program. This worsening of extrapyramidal symptoms can be attributed to an increase in blood flow to the site of injection during exercise, accelerating the rate of absorption and bioavailability of the antipsychotic medication and subsequently increasing dopamine D_2_ receptor blockade. When monitoring drug therapy for patients receiving long-acting injectable antipsychotic medications, pharmacists and other healthcare professionals need to consider exercise as a contributing factor for the emergence of extrapyramidal symptoms.

## 1. Introduction

Schizophrenia is a severe psychiatric disorder that has a lifetime prevalence of approximately 0.87% [1]. Second-generation antipsychotic medications have been recommended as the first-line pharmacological treatment for schizophrenia and a range of other psychotic disorders [2]. These medications are available as oral and, for many, also as long-acting injectable (LAI) formulations. LAI antipsychotic medications are formulated to facilitate the sustained release of the deposited antipsychotic over a number of weeks, thus increasing the dosing interval required for the maintenance of therapeutic plasma concentrations under steady-state conditions [3]. As a result, LAI antipsychotics may be superior to oral antipsychotics in terms of adherence, preventing relapse, and reducing the risk of rehospitalization [4].

Apart from the site of injection, the absorption and bioavailability of LAI antipsychotic medications into the vasculature are also dependent on physiological factors that include obesity and local blood flow [5,6,7]. With regard to the latter, exercise increases blood flow to active muscles, thereby increasing the rate of absorption of the deposited drug. This phenomenon has been well documented in patients with diabetes, where exercise increased the absorption of the injected insulin and thereby increased the risk of iatrogenic hypoglycemia [8]. This is also relevant for patients with schizophrenia as exercise intervention is increasingly investigated and recommended as an adjunct to antipsychotic treatment due to its favorable effects on the positive and negative symptoms of schizophrenia [9,10], stress and anxiety [11], concentration and attention [12], metabolic syndrome [13,14], and hippocampal neurogenesis [15,16]. Similar to the case of diabetic patients, exercise can increase the rate of absorption of the deposited antipsychotic, and as a result, there can be emergent adverse effects that are dose-related, including cardiovascular and metabolic abnormalities, hyperprolactinemia, and extrapyramidal symptoms (EPS) [17,18,19]. In this study, we report two cases of exercise-associated worsening of EPS in participants with schizoaffective disorder who had been stabilized on paliperidone palmitate LAI (PP1M) and entered a randomized controlled 12-week exercise study.

## 2. Case Presentation

The aim of the exercise study, known as the Psychosis, Exercise, and Hippocampal Plasticity (PEHP) Study, was to determine the effects of exercise on brain morphology and health in 27 individuals with chronic schizophrenia and is described in detail elsewhere [20,21]. Each consenting participant was randomly assigned to either a 12-week aerobic or resistance exercise program and completed three 30-min exercise sessions per week at moderate intensity. The Extrapyramidal Symptom Rating Scale (ESRS) was used to rate the severity of EPS at baseline, week 6, and week 12 [22]. This study was approved by the University of British Columbia’s Clinical Research Ethics Board and is registered with ClinicalTrials.gov (NCT01392885). In total, there were six participants treated with LAI antipsychotic medications at study entry. Three of these participants discontinued their LAI antipsychotics prior to week 6 and one participant was prescribed oral risperidone at week 6, which may have confounded any increase in his/her EPS. Conversely, the other two participants had been stabilized on their LAI antipsychotics for more than one year prior to entering the study, continued their LAI antipsychotics at stable doses throughout the study, and thus any increases in their EPS were not confounded by antipsychotic polypharmacy. We discuss here the latter two participants who were stabilized on LAI antipsychotic medications and exhibited worsening of EPS throughout the exercise program.

Patient A was a 28-year-old Caucasian man diagnosed with schizoaffective disorder and was randomly assigned to the 12-week aerobic exercise program. His baseline treatment regimen included a maintenance dose of PP1M 150 mg (deltoid) administered every 4 weeks, olanzapine 15 mg/d, and benztropine 3 mg/d. The dose of PP1M remained unchanged throughout the program, whereas the dose of olanzapine was reduced to 10 mg/d by week 6 and to 7.5 mg/d by week 12 in an attempt to eliminate antipsychotic polypharmacy. Additionally, benztropine 3 mg/d was switched to procyclidine 15 mg/d at week 7. The total ESRS score changed from 28 at baseline to 55 at week 6, and 41 at week 12. The ESRS Clinical Global Impression of Severity (CGI-S) of both dyskinesia and parkinsonism changed from 2 (very mild) at baseline to 4 (moderate) at week 6, and 3 (mild) at week 12. The severity of akathisia changed from 0 (none) at baseline to 1 (looks restless, nervous, impatient, uncomfortable) at week 6, and 3 (often needs to move one extremity or to change position) at week 12. Dystonia was absent throughout the exercise program.

Patient B was a 40-year-old Haitian man diagnosed with schizoaffective disorder and was randomly assigned to the 12-week resistance exercise program. His baseline treatment regimen included a maintenance dose of PP1M 150 mg (deltoid) administered every 4 weeks, quetiapine 600 mg/d, and benztropine 4 mg/d, all of which remained unchanged throughout the program. The total ESRS score changed from 30 at baseline to 46 at week 6, and 69 at week 12. The ESRS CGI-S of dyskinesia changed from 0 (absent) at baseline to 4 (moderate) at week 6, and 3 (mild) at week 12. The ESRS CGI-S of parkinsonism changed from 3 (mild) at baseline and week 6, to 5 (moderately severe) at week 12. Akathisia and dystonia were absent throughout the exercise program.

## 3. Discussion

Physiological factors, such as local blood flow and exercise, can significantly affect the rate and extent of absorption of intramuscularly administered drugs [5,6,7]. Drugs administered into the deltoid muscle have a faster rate of absorption than those administered into the gluteal muscle because of greater blood flow [6,7]. Exercise increases blood flow from other organs to active muscles and produces heat that may facilitate the diffusion of the deposited drug [5]. Our cases are very consistent with another case reported previously [23]. Seeman (2016) noted complaints of EPS from one of her patients specifically during the months the patient played soccer [23]. Her patient was receiving fluphenazine enanthate depot administered into the gluteal muscle every 3 weeks. Seeman attributed this event to an exercise-associated increase in blood flow and heat at the injection site, which increased the drug’s rate of absorption and bioavailability, resulting in more dopamine D_2_ blockade and thus EPS [23]. Taken together, several observations can be made: (1) both types of exercise (aerobic and resistance) may be associated with worsening of EPS in patients receiving LAI antipsychotic medications, (2) an exercise-associated increase in blood flow may affect the pharmacokinetics of LAI antipsychotic medications at both sites of injection (deltoid and gluteal muscles), and (3) exercise-associated EPS may be relevant to all LAI antipsychotic medications.

The concomitant use of other oral antipsychotic medications may appear to confound our cases. However, the oral antipsychotics used by our patients were olanzapine and quetiapine, which have lower EPS risks relative to other antipsychotics, including paliperidone [24]. Additionally, in Patient A, the severity of EPS increased from baseline to week 6 despite the decrease in the dose of olanzapine. Exercise is not believed to have a substantial effect on the absorption of orally administered drugs [5]. If anything, exercise redistributes blood away from the gastrointestinal tract towards the active muscles, reducing the splanchnic-hepatic blood flow and subsequent absorption of orally administered drugs [5]. In this regard, other participants in the study who were maintained solely on oral antipsychotic medications for the duration of the exercise program did not experience worsening of EPS, even with increases in dose. In some cases, EPS actually improved. For example, a 27-year-old male who was randomly assigned to the aerobic exercise program and treated with olanzapine that was increased from 20 mg/d at baseline to 45 mg/d at week 6 demonstrated a reduction in total ESRS score from 44 at baseline to 27 at week 6. Similarly, a 41-year-old male who was randomly assigned to the resistance exercise program and treated with haloperidol 10 mg/d throughout the exercise program and olanzapine that increased from 10 mg/d at baseline to 30 mg/d at week 12 demonstrated a reduction in total ESRS score from 27 at baseline to 19 at week 12.

Patient A’s reduced EPS from week 6 to 12 could have been, in part, due to changes in his medications (i.e., reducing the dose of olanzapine and adding procyclidine in place of benztropine). However, it is noteworthy that his EPS markedly worsened over the first 6 weeks of aerobic exercise despite the reduced olanzapine dose and required a switch to a different anticholinergic agent. Although a slight improvement in EPS was noticed from week 6 to 12, the severity of his akathisia continued to worsen. In the case of Patient B, the association between exercise and EPS was much stronger as his EPS progressively worsened with no confounding changes in his pharmacological treatment. Also worth noting is the fact that both participants were on PP1M for more than one year prior to the study and as such had already reached steady-state concentrations. Thus, no temporal relationship can be made between the worsening of EPS in our cases and peak plasma concentrations of paliperidone associated with reaching a steady state.

## 4. Conclusions

In conclusion, these cases highlight the importance for pharmacists and other healthcare professionals to consider the impact of non-drug factors, such as exercise, as a contributing cause of adverse events in patients treated with LAI antipsychotic medications. In particular, if a patient unexpectedly develops EPS after being stabilized on an LAI antipsychotic medication, pharmacists and other healthcare professionals should ask the patient about any recent engagement in lifestyle interventions, including exercise, or for that matter, physical labour. This is especially important as exercise intervention is increasingly utilized as an adjunct to antipsychotic treatment in individuals with schizophrenia [9,10,16]. Future investigations should be placebo-controlled, assess various levels of exercise intensity, and incorporate antipsychotic medication plasma concentrations in the protocol.

## Data Availability

Not applicable.

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
