# Peer review of "Exercise and Worsening of Extrapyramidal Symptoms during Treatment with Long-Acting Injectable Antipsychotics"

_pharmacy, 2021, doi:10.3390/pharmacy9030123_

Round 1
Reviewer 1 Report
On the whole, I read this report with some interest but I found certain areas rather misleading and skewed.
Specific comments:
- Suggest changing the title to "Exercise and worsening of extrapyramidal symptoms ... ".
- Please change "Second-generation antipsychotic medications can effectively treat schizophrenia" to "Second-generation antipsychotic medications are used to treat schizophrenia".
- The general health benefits of exercise, be it a short-bout or long-term should be more clearly stated in the introduction section. For example, exercise has been linked to increased blood flow to the brain and neurotransmitter levels, enhanced plasticity and better focus, attention and information processing in typically-developing children and children with attention-deficit/hyperactivity disorder (citation: pubmed.ncbi.nlm.nih.gov/28917364).
- Understand that the two subjects were taken from a larger cohort study, known as the Psychosis, Exercise, and Hippocampal 55 Plasticity (PEHP) Study. However, it is unclear how the authors decided to report specifically on these two cases. As part of the larger study cohort, did only two subjects experience EPS? This seems like a rather biased and skewed portrayal otherwise.
- What is the clinical strategy for these patients who had intolerable EPS? The authors may add that despite the introduction of newer antipsychotic medications, the management of schizophrenia remains a substantial clinical challenge. When faced with a patient that does not respond to or cannot tolerate standard antipsychotic therapy, physicians have limited treatment strategies (citation: pubmed.ncbi.nlm.nih.gov/32728915). It does not help that a significant proportion of patients with schizophrenia do not respond sufficiently to antipsychotics and combination therapy is often required. For patients with inadequate response in target psychotic or schizophrenia symptoms, adding a second antipsychotic medication has yielded largely negative results based on randomized trials (citation: pubmed.ncbi.nlm.nih.gov/9519099).
- Important considerations to ensure valid and reliable results when ordering antipsychotic plasma levels include 1) availability of clinically validated assays for immediate-release formulations and, if available, for extended-release formulations as well; 2) ordering the test after the drug has achieved steady state, usually at least 5 drug half-lives; 3) testing at the recommended sampling time; although a random level is adequate when nonadherence is suspected, a trough level is preferable, particularly for drugs with short half-lives or to rule out rapid metabolism; and 4) informing the laboratory of the possibility of inadequate adherence and whether the drug is immediate release or extended release. There are different for different drugs and would also add to the cost of treatment, hence it is unlikely (and should not) be applied to all patients on antipsychotic medications. The discussion on the monitoring of antipsychotic drug plasma levels should be more nuanced.
- "... such as exercise, as a contributing cause of adverse events in patients" - how common is this? I suggest authors include a review of reported cases in the discussion section. Moreover, it would be useful to discuss if this is likely to dose-dependent, i.e. more likely to happen with higher intensity training?
Author Response
Reviewer 1: Comments and Suggestions for Authors
On the whole, I read this report with some interest but I found certain areas rather misleading and skewed.
Specific comments:
1) Suggest changing the title to "Exercise and worsening of extrapyramidal symptoms ... ".
Our Response: We have changed the title accordingly.
2) Please change "Second-generation antipsychotic medications can effectively treat schizophrenia" to "Second-generation antipsychotic medications are used to treat schizophrenia".
Our Response: We have changed the first sentence of the abstract as follows (lines 13-15): “Second-generation antipsychotic medications are used to treat schizophrenia and a range of other psychotic disorders, although adverse effects, including cardiovascular and metabolic abnormalities and extrapyramidal symptoms, are often inevitable.”
3) The general health benefits of exercise, be it a short-bout or long-term should be more clearly stated in the introduction section. For example, exercise has been linked to increased blood flow to the brain and neurotransmitter levels, enhanced plasticity and better focus, attention and information processing in typically-developing children and children with attention-deficit/hyperactivity disorder (citation: pubmed.ncbi.nlm.nih.gov/28917364).
Our Response: Rather than the “general” health benefits of exercise, we focused on the health benefits of exercise for schizophrenia, which is more relevant to our study. The following sentences were added (lines 48-54): “This is also relevant for patients with schizophrenia as exercise intervention is increasingly investigated and recommended as an adjunct to antipsychotic treatment due to its favorable effects on the positive and negative symptoms of schizophrenia [9,10], stress and anxiety [11], concentration and attention [12], metabolic syndrome [13,14], and hippocampal neurogenesis [15,16]. Similar to the case of diabetic patients, exercise can increase the rate of absorption of the deposited antipsychotic, and as a result, there can be emergent adverse effects that are dose-related, including…”
4) Understand that the two subjects were taken from a larger cohort study, known as the Psychosis, Exercise, and Hippocampal 55 Plasticity (PEHP) Study. However, it is unclear how the authors decided to report specifically on these two cases. As part of the larger study cohort, did only two subjects experience EPS? This seems like a rather biased and skewed portrayal otherwise.
Our Response: In total, there were 6 subjects on long-acting injectable antipsychotics enrolled in the PEHP study. We have added the following (lines 68-77): “In total, there were 6 participants treated with LAI antipsychotic medications at study entry. Three of these participants discontinued their LAI antipsychotics prior to week 6 and one participant was prescribed oral risperidone at week 6, which may have confounded any increase in his/her EPS. Conversely, the other two participants had been stabilized on their LAI antipsychotics for more than one year prior to entering the study, continued their LAI antipsychotics at stable doses throughout the study, and thus any increases in their EPS were not confounded by antipsychotic polypharmacy. We discuss here the latter two participants who were stabilized on LAI antipsychotic medications and exhibited worsening of EPS throughout the exercise program.”
5) What is the clinical strategy for these patients who had intolerable EPS? The authors may add that despite the introduction of newer antipsychotic medications, the management of schizophrenia remains a substantial clinical challenge. When faced with a patient that does not respond to or cannot tolerate standard antipsychotic therapy, physicians have limited treatment strategies (citation: pubmed.ncbi.nlm.nih.gov/32728915). It does not help that a significant proportion of patients with schizophrenia do not respond sufficiently to antipsychotics and combination therapy is often required. For patients with inadequate response in target psychotic or schizophrenia symptoms, adding a second antipsychotic medication has yielded largely negative results based on randomized trials (citation: pubmed.ncbi.nlm.nih.gov/9519099).
Our Response: We fail to understand Reviewer’s comment in the context of our manuscript. Firstly, the Reviewer asks, “what is the clinical strategy for these patients who had intolerable EPS?” Our two patients in this case do not have “intolerable” EPS. Secondly, the treatment strategy of “intolerable” ESP is out of scope for this case report. The focus of this manuscript is to report how exercise may worsen EPS not how EPS are treated. We have not made any change to the manuscript regarding this point.
The Reviewer then changes his/her focus to treatment of target symptoms, inadequate response, and antipsychotic polypharmacy. Respectfully, this manuscript is not about the “management” of patients with “inadequate response” with strategies that may involve the employment of “adding a second antipsychotic”. We have not made any change to the manuscript regarding this point.
6) Important considerations to ensure valid and reliable results when ordering antipsychotic plasma levels include 1) availability of clinically validated assays for immediate-release formulations and, if available, for extended-release formulations as well; 2) ordering the test after the drug has achieved steady state, usually at least 5 drug half-lives; 3) testing at the recommended sampling time; although a random level is adequate when nonadherence is suspected, a trough level is preferable, particularly for drugs with short half-lives or to rule out rapid metabolism; and 4) informing the laboratory of the possibility of inadequate adherence and whether the drug is immediate release or extended release. There are different for different drugs and would also add to the cost of treatment, hence it is unlikely (and should not) be applied to all patients on antipsychotic medications. The discussion on the monitoring of antipsychotic drug plasma levels should be more nuanced.
Our Response: The focus of our report is not about the “nuances” of therapeutic drug monitoring using “plasma levels”. From my decades of clinical experience, I have never once ordered/recommended obtaining plasma concentrations of an antipsychotic because a patient had developed EPS. This report is based on a research protocol in which antipsychotic plasma concentrations were not part of the design. In this regard, we have mentioned this as a potential limitation in the very last sentence of the conclusion (lines 157-158) as follows: “Future investigations should be placebo-controlled, assess various levels of exercise intensity, and incorporate antipsychotic medication plasma concentrations in the protocol.”
7) "... such as exercise, as a contributing cause of adverse events in patients" - how common is this? I suggest authors include a review of reported cases in the discussion section. Moreover, it would be useful to discuss if this is likely to dose-dependent, i.e. more likely to happen with higher intensity training?
Our Response: By definition, case reports describe an unusual, rare, or novel occurrence. When we prepared this manuscript, we did a search of the literature and referenced the only other case that we could find. We already mentioned this other case as follows (lines 106-109): “Our cases are very consistent with another case reported previously [23]. Seeman (2016) noted complaints of EPS from one of her patients specifically during the months the patient played soccer [23]. Her patient was receiving fluphenazine enanthate depot administered into the gluteal muscle every 3 weeks.”
The Reviewer suggests that it “would be useful to discuss if this is likely dose-dependent, i.e., more likely to happen with higher intensity training?” Once again, case reports are rare occurrences so we can only speculate whether EPS would worsen with “higher intensity training”. However, we do believe this is an important question and we have modified that last sentence of the conclusion as follows (lines 157-159): “Future investigations should be placebo-controlled, assess various levels of exercise intensity, and incorporate antipsychotic medication plasma concentrations in the protocol.”
Reviewer 2 Report
Interesting case report on the exercise and extrapyramidal symptoms during treatment with long-acting injectable antipsychotics.
Comments
1) PEHP study. How many participants were enrolled in the study? What were the inclusion and exclusion criteria?
2) The two participants developed EPS symptoms in the cases reported. How many participants on LAIs did not develop EPS symptoms?
Please expand the introduction and discussion and include 3-5 additional references.
Author Response
Reviewer 2: Comments and Suggestions for Authors
Interesting case report on the exercise and extrapyramidal symptoms during treatment with long-acting injectable antipsychotics.
Comments
1) PEHP study. How many participants were enrolled in the study? What were the inclusion and exclusion criteria?
Our Response: The number of participants in the PEHP study was 27. We have now added this as follows (lines 61-63): “Plasticity (PEHP) Study, was to determine the effects of exercise on brain morphology and health in 27 individuals with chronic schizophrenia and is described in detail elsewhere [20,21].”
We did not add the inclusion and exclusion criteria to this manuscript as requested since it would take up considerable space (please see Inclusion/Exclusion criteria list below). However, we have provided the reference to the study (i.e., reference #20) if the reader wishes to examine the details of the protocol.
Inclusion Criteria: 1) Age 21 to 45, 2) Able to provide written, informed consent in English, 3) Patients may be on prescribed medications, 4) DSM-IV (Diagnostic and Statistical Manual of Mental Disorders, Fourth Edition) diagnosis of schizophrenia or schizoaffective disorder, 5) Normal visual acuity (or normal visual acuity achievable with corrective lenses), 6) Physical ability to engage in a regular exercise program (as determined by their treating physician and the completion of the ePARmed-X+).
Exclusion Criteria: 1) A history of developmental disorders (e.g., autism, mental retardation, Down’s Syndrome), 2) A current DSM-IV diagnosis of substance dependence (during prior 120 days, excluding tobacco), 3) Any history of DSM-IV diagnoses (Axis I) for other psychiatric disorders, 4) History of angina, heart attack, or transient ischemic attacks, 5) Non-independent mobility of limb prostheses, 6) A history of organic disorders or severe head trauma (e.g., dementia or head injury leading to loss of consciousness for > 5 mins), 7) Contraindications for neuroimaging (metal implants, non-removable orthodontic devices, severe claustrophobia, pregnancy, or surgeries within the previous six months), 8) Already enrolled currently in a regular exercise program, 9) Currently not stable on medications.
2) The two participants developed EPS symptoms in the cases reported. How many participants on LAIs did not develop EPS symptoms?
Our Response: In total, there were 6 subjects on long-acting injectable antipsychotics enrolled in the PEHP study. We have added the following (lines 68-77): “In total, there were 6 participants treated with LAI antipsychotic medications at study entry. Three of these participants discontinued their LAI antipsychotics prior to week 6 and one participant was prescribed oral risperidone at week 6, which may have confounded any increase in his/her EPS. Conversely, the other two participants had been stabilized on their LAI antipsychotics for more than one year prior to entering the study, continued their LAI antipsychotics at stable doses throughout the study, and thus any increases in their EPS were not confounded by antipsychotic polypharmacy. We discuss here the latter two participants who were stabilized on LAI antipsychotic medications and exhibited worsening of EPS throughout the exercise program.”
3) Please expand the introduction and discussion and include 3-5 additional references.
Our Response: We added the following (lines 48-54): “This is also relevant for patients with schizophrenia as exercise intervention is increasingly investigated and recommended as an adjunct to antipsychotic treatment due to its favorable effects on the positive and negative symptoms of schizophrenia [9,10], stress and anxiety [11], concentration and attention [12], metabolic syndrome [13,14], and hippocampal neurogenesis [15,16]. Similar to the case of diabetic patients, exercise can increase the rate of absorption of the deposited antipsychotic, and as a result, there can be emergent adverse effects that are dose-related, including…”
Reviewer 3 Report
Overall a very interesting and important cases to keep in mind for patients being treated with long-acting injectable antipsychotics.
Line 32: it may be more inclusive to use "psychiatric" disorder vs. "mental" disorder
Line 39-41: based on the results and analysis done in reference 4, it is more appropriate to say "LAI antipsychotics may be superior to oral antipsychotics"
Was patient A's olanzapine decreased due to the worsening EPS or for other reasons?
Is there any data that could give us an indication of how quickly EPS worsened after receiving the injection during this period of exercise? Would be helpful to know severity of EPS during that 6 weeks
When did patients receive their injections relative to the EPS assessment? For example, were they are baseline and every 4 weeks after that? Or on a different schedule during the 12 weeks of exercise?
Author Response
Reviewer 3: Comments and Suggestions for Authors
Overall a very interesting and important cases to keep in mind for patients being treated with long-acting injectable antipsychotics.
1) Line 32: it may be more inclusive to use "psychiatric" disorder vs. "mental" disorder
Our Response: We have changed the sentence as follows (lines 32-33): “Schizophrenia is a severe psychiatric disorder that has a lifetime prevalence of approximately 0.87% [1].”
2) Line 39-41: based on the results and analysis done in reference 4, it is more appropriate to say "LAI antipsychotics may be superior to oral antipsychotics"
Our Response: We have changed the sentence as follows (lines 39-41): “As a result, LAI antipsychotics may be superior to oral antipsychotics in terms of adherence, preventing relapse, and reducing the risk of rehospitalization [4].”
3) Was patient A's olanzapine decreased due to the worsening EPS or for other reasons?
Is there any data that could give us an indication of how quickly EPS worsened after receiving the injection during this period of exercise? Would be helpful to know severity of EPS during that 6 weeks. When did patients receive their injections relative to the EPS assessment? For example, were they are baseline and every 4 weeks after that? Or on a different schedule during the 12 weeks of exercise?
Our Response: The dose of olanzapine was reduced in an attempt to eliminate antipsychotic polypharmacy. We have added the following sentence as follows (lines 81-84): “The dose of PP1M remained unchanged throughout the program, whereas the dose of olanzapine was reduced to 10 mg/d by week 6 and to 7.5 mg/d by week 12 in an attempt to eliminate antipsychotic polypharmacy.”
The Reviewer brings up a very good point about “how quickly EPS worsened after receiving the injection during this period of exercise”. As with clinical trials, the research protocol had defined times in which patients were assessed. In this case, patients were rated for EPS at baseline, Week 6, and Week 12. As such, we cannot comment with certainty about the temporal relationship between the time of the injection and the onset of EPS. We did not make any change to the manuscript regarding this point.
Another good point raised is about the temporal relationship between the day of injection and the assessment of EPS. Upon review of the research database, there is no mention of the date when the long-acting injectable antipsychotic was given. The database only reports when a particular medication was started, the dose administered, and the frequency. Thus, although we know the participants received 150 mg of PP1M every four weeks, we do not know the exact dates the injections were given. To determine the exact date of administration of the PP1M would require us to order the chart from the Department of Clinical Records. Unfortunately, we are unable to do this before our response to the Reviewers’ Comments for this manuscript was due. If this Reviewer feels that this information is important in explaining our findings, we will ask the Editor for an extension. However, in our opinion, we do not believe that this information would significantly change the premise of this manuscript.
Round 2
Reviewer 1 Report
Thanks for addressing my concerns in a comprehensive manner.